# Muscle Activation Frequency Relationship with Cost of Transport at Increasing Walking Speed in Preliminary Study Reveals Interplay of Both Active and Passive Dynamics

Grace Van Namen, David Adair, Dean J Calsbeek and Rajat Emanuel Singh *

Department of Kinesiology, Northwestern College, Orange City, IW 51041, USA; gvannamen@gmail.com (G.V.N.); david.adair@nwciowa.edu (D.A.); dcalsbeek@nwciowa.edu (D.J.C.)
* Correspondence: rajat.singh@nwciowa.edu

**Abstract:** Metabolic cost plays a critical role in gait biomechanics, particularly in rehabilitation. Several factors influence metabolic cost during walking. Therefore, this study aimed to examine the relationship between metabolic cost and muscle activity, focusing on the frequency of EMG signals during walking. We recruited nine participants (five male and four female, age range 20–48 years) who walked for four minutes at six different speeds (ranging from 1.8 to 5.9 mph). EMG data were collected from the dominant lower leg muscles, specifically the lateral gastrocnemius (GAS-L) and the anterior tibialis (AT). Oxygen respiration was measured using open-circuit spirometry. Energy expenditure was estimated as the cost of transport (COT). The EMG data were analyzed using frequency domain features, such as the area under the curve of power spectral density (PSD-AUC) and the maximal distance between two points before and after the peak of the power spectral density curve (MDPSD). The results indicated that PSD-AUC is a better measure than MDPSD for understanding the relationship between activation frequency and COT. PSD-AUC demonstrated an increasing curvilinear trend ($R^2$ = 0.93 and 0.77, second order polynomial fit), but the AT displayed higher variability. MDPSD exhibited more nonlinearity ($R^2$ = 0.17–28, second order polynomial fit), but MDPSD demonstrated statistically significant differences ($p < 0.05$, $t$-test independent) in frequency between the GAS-L (64–237 Hz) and AT (114–287 Hz) during slow walking. Additionally, the relationship between COT and PSD-AUC revealed a U-shaped curve, suggesting that high COT is a function of both active and passive dynamics during walking. These findings will be valuable in rehabilitating individuals who suffer from gait-related disorders, especially those related to muscle inefficiency.

**Keywords:** cost of transport; muscle activation; active dynamics; passive dynamics

## 1. Introduction

In the field of human biomechanics, the metabolic cost refers to the energy expended by the human body to perform a movement. Various methods are available for computing metabolic costs, utilizing principles of direct and indirect calorimetry. The metabolic cost can also be estimated through the use of musculoskeletal model simulations. Models such as the Hill-type model, Huxley cross-bridge model, and joint space numerical model are employed for this purpose [1]. Indirect calorimetric measurements are generally used as a reference to account for the precision and performance of simulation-based methods. Therefore, indirect calorimetric measures seem to be a more effective means of estimating metabolic costs.

One limitation of indirect calorimetric instrumentation, such as open-circuit spirometry, is the lack of information available regarding variables such as muscle activity and joint angles. It is important to consider variables such as muscle and joint mechanics, as they directly influence energy expenditure. To overcome this limitation, gas exchange

data are typically collected while simultaneously measuring muscle activity and joint kinematics through biomechanics instrumentation. These additional measurements provide a foundation and rationale for understanding changes in metabolic activity.

Several researchers have studied the regulation of metabolic cost during different mechanical tasks by measuring oxygen and carbon dioxide exchange using a metabolic cart with biomechanical instruments [2,3]. A specific area of focus that has attracted attention from researchers investigating the management of metabolic costs is the study of gait. This is because walking alone accounts for approximately 30% of energy expenditure, and the metabolic cost of walking is optimized to meet the demands of different walking conditions. Previous studies have demonstrated that humans tend to choose a range of gait parameters, such as stride length and stride rate, to minimize energy expenditure and adopt a more efficient gait [4,5]. For instance, the average walking speed, which is also referred to as self-selected pace, represents the most metabolically efficient pace. Therefore, the relationship between incrementing walking speed (ranging from slow to fast) and metabolic cost appears to follow a U-shaped curve, with the vertex of the U indicating the most efficient region for walking [6]. These inherent features render investigations surrounding gait of significant interest among researchers.

In gait efficiency studies, researchers commonly measure oxygen uptake and electromyography (EMG) during graded walking exercises [7,8]. This approach is utilized because muscles actively metabolize macronutrients to produce adenosine triphosphate (ATP) in the presence of oxygen, while also releasing carbon dioxide during strenuous exercise. Therefore, the analysis of EMG data while walking at incrementing speed levels offers valuable biomarkers that aid in identifying optimal gait parameters and can help in developing effective rehabilitation devices and therapies. These optimal parameters are crucial for individuals with gait abnormalities and asymmetries, such as amputees. For example, an amputee using prostheses tends to walk slower and experience higher energy expenditure compared to non-amputees due to the mass distribution of the prosthesis device [9]. Hence, by analyzing EMG activity and metabolic cost at varying speeds, it is possible to identify gait conditions where muscles experience significant neuromuscular burden due to biomechanical constraints, leading to high energy expenditure.

The current research literature regarding quantifying EMG activation has certain limitations because it primarily focuses on utilizing the time domain feature at a fixed speed, such as the amplitude of EMG activity [10–12]. These studies generally account for metabolic cost as the cost of transport (COT) and determine its relationship with the time domain features of EMG patterns [13]. However, it is crucial to acknowledge that frequency domain analysis, specifically power spectrum analysis, is a more suitable index for capturing motor unit recruitment patterns of muscles, as demonstrated in previous studies [14,15]. As participants increase their speed, they recruit a diverse range of muscle fibers with varying firing frequencies. This encompasses both slow twitch and fast twitch fibers. Using frequency domain analysis allows for a comprehensive assessment of their recruitment patterns. Therefore, for a study that emphasizes the measurement of gait efficiency, employing frequency domain analysis of EMG with COT would be a more appropriate measure to study gait efficiency. In this study, power spectral density (PSD) was employed to compute muscle activation frequency, and the relationship between muscle activation frequency and COT at incrementing speeds was studied.

The main aim of this study was to understand changes in muscle activation frequency as walking speed increases and identify its relationship with metabolic cost. We hypothesize that walking at incrementing speed levels will have a nonlinear impact on muscle activation frequencies due to an interplay of both active and passive dynamics. To test this hypothesis, we first studied the changes in PSD at different speed levels using measures such as the area under the curve (PSD-AUC) and maximal distance between data points on a power spectrum curve before and after peak value (MDPSD). Subsequently, the relationship between these measures and the metabolic cost at increasing speed levels was examined. These analyses provide insights into efficient gait and the required muscle contraction

patterns that facilitate such efficiency. As a result, we anticipate that the findings from this study will provide valuable insights for rehabilitating patients with gait abnormalities, aiding in their recovery and improvement.

## 2. Materials and Methods

The institutional review board of Northwestern College, Orange City approved this study. All the participants recruited in this study gave their consent by signing a consent form. Individuals with movement disorders were not recruited in this study. The participants in this study were neither overweight nor obese, and they exhibited no signs of metabolic syndrome, infection, or fatigue on the assessment day. Additionally, participants did not consume caffeine or prescription-based medicine on the assessment day.

### 2.1. Participant Recruitment and Description

We determined the sample size for the trials/gait cycles using a power analysis test. A number of 18 trials/gait cycles was sufficient to explain a power of 0.8, significance level of 0.5, and a Cohen value of 0.4 at each speed level. Nine participants were recruited ranging from 20 to 48 $\pm$ 10.1 years old. The participant group included four females and five males. The average participant height was 178.48 $\pm$ 6.75 cm, and the average participant weight was 81.77 $\pm$ 12.93 kg. The demographic data are shown in Table 1 below.

**Table 1.** Demographic Data.

| S. No. | Age (Years) | Height (cm) | Weight (kg) | Avg. Speed (mph) |
|--------|-------------|-------------|-------------|------------------|
| P1 | 37 | 171.6 | 65.9 | 3.9 |
| P2 | 20 | 184.4 | 80 | 3.4 |
| P3 | 48 | 185.8 | 100 | 2.9 |
| P4 | 21 | 180.2 | 95.5 | 3.3 |
| P5 | 27 | 177.4 | 73.1 | 3.1 |
| P6 | 38 | 188.9 | 84.9 | 3.8 |
| P7 | 22 | 173.4 | 95 | 3.6 |
| P8 | 23 | 174.9 | 75 | 3.1 |
| P9 | 20 | 169.7 | 66.1 | 2.8 |

### 2.2. Experimental Protocol

There were two sessions conducted for this study, across two days. The first session consisted of measuring the anthropometric data (height (Health o meter, McCook, IL, USA) and weight (Detecto, Webb City, MI, USA)) and determining the average walking speed of each individual. A 10 m walking test was used to determine the average walking speed [16] of each participant. Two markers were placed ten meters apart overground to indicate where participants should start and stop walking. The time to walk the ten meters was measured in seconds and used to calculate speed. This task was repeated three times, and the results were averaged to determine the average walking speed. The average walking speed among the nine participants was recorded to be 3.32 $\pm$ 0.39 mph. Upon recording the average walking speed, six additional levels of walking speeds for an individual were also determined for the experiment. A value of 0.5 mph was cumulatively added and subtracted from the average walking speed to determine the different walking speed levels for eight out of nine participants, as shown in Table 2. For one of the nine participants, two below-average walking speed levels were determined based on his/her comfort level.

During the second day of data collection, the assessment included metabolic measurements during walking at incrementing speeds and EMG recording. At the start of the session, participants performed a self-selected warm-up walk on a treadmill. Two minutes after the warmup session, the second-day experimental protocol was started, which involved participants walking at increasing speed levels (at 0% incline (grade) on the treadmill for all levels). Each walking speed level lasted four minutes to ensure the participant would reach a metabolic steady state. A one-minute rest phase occurred between stages to

reduce fatigue, and the rate of perceived exertion was recorded for each stage using the Borg scale [17]. Eight out of the nine participants performed seven levels of walking speed, whereas one out of the nine participants performed five levels. At the time of completion of the seven levels, participants completed the assessment with a short recovery stage.

**Table 2.** Participants walking speed levels.

| S. No. | Avg. Speed (mph) | Speed Levels (mph) |
| --- | --- | --- |
| P1 | 3.9 | 2.9, 3.4, 4.4, 4.9, 5.4, 5.9 |
| P2 | 3.4 | 2.4, 2.9, 3.9, 4.4, 4.9, 5.4 |
| P3 | 2.9 | 1.9, 2.4, 3.4, 3.9, 4.4, 5.0 |
| P4 | 3.3 | 2.3, 2.8, 3.8, 4.3, 4.8, 5.3 |
| P5 | 3.1 | 2.1, 2.3, 3.6, 4.1, 4.6, 5.1 |
| P6 | 3.8 | 2.8, 3.3, 4.3, 4.8, 5.3, 5.8 |
| P7 | 3.6 | 2.6, 3.1, 4.1, 4.6, 5.1, 5.6 |
| P8 | 3.1 | 2.1, 2.6, 3.6, 4.1, 4.6, 5.1 |
| P9 | 2.8 | 1.8, 2.3, 3.3, 4.8, 4.3, 4.8 |

### 2.3. Metabolic Data Recording

The COSMED Quark CPET (COSMED Sri, Rome, Italy) metabolic cart was used for data collection. At each speed level, gas exchange was recorded for four minutes on a breath-by-breath basis. The metabolic cart equipped with the OMNIA software system (Version 2.1.1) (COSMED Sri, Rome, Italy) provided the data with steady-state $VO_2$ plots in real time, which assisted in time stamping where a steady state was being reached at each speed level. During the one minute of rest between the different speed levels, no metabolic data was recorded. Upon completion of the protocol, the raw data was smoothed with 10 s time averages.

### 2.4. Gross Cost of Transport Calculation

To determine the efficiency of walking at different speed levels, a customized code on MATLAB 2017 was used to calculate the gross cost of transport (COT) from $VO_2$ data. Additionally, it was difficult to differentiate between walking faster and running. To address this issue, we employed the COT as a measure to identify the speed level at which participants transitioned from walking to running during the protocol. Any speed level where participants demonstrated running, as indicated by COT, was excluded from further analysis in our experiment. We used the maximal $VO_2$ data point (steady state = $\dot{V}O_2$) recorded at each speed level and divided that sample with the respective velocity (V), as per Equation (1) [18].

$$\text{Gross COT} = \frac{\dot{V}O_2}{V} \tag{1}$$

### 2.5. EMG Recording

A Wireless Delsys Trigno+ research-based EMG system was used to record muscle activity. For the EMG recording, participants were asked not to wear any lotions or oils prior to arriving for the second session. The surface of the skin during the second session was cleaned with alcohol wipes to rid the skin of any sweat or naturally produced oils. This allowed for the EMG sensors to adhere and stick to the skin properly and provide a cleaner signal [19]. The lateral gastrocnemius (GAS-L) and anterior tibialis muscles (AT) muscles of the dominant leg were identified by asking participants to perform plantar flexion and dorsiflexion. After muscle identification, two EMG sensors were placed on the surface of each muscle belly to record the muscle activity during the experiment. Additional athletic tape was wrapped around the lower leg to secure the sensor's placement. Then, EMG data were recorded during each level of walking continuously for four minutes at a sampling rate of 2341 samples per second. The EMG data collection took place at the

same time as the metabolic data collection, as verbal cues synchronized the start and stop of data collection.

### 2.6. EMG Processing

The EMG data collected (2 channels × 568,140 samples) were then processed through the EMG Works analysis software system (Version 4.7.3.0). First, the data were processed by running an IIR band pass filter with a cutoff frequency of 20–500 Hz to remove noise from the EMG data [20]. The DC offset was removed from the EMG signal, and then the power spectral density (PSD) was calculated from the filtered data, with the parameters including a window type, length, and overlap. Other features from the PSD plot, such as the area under the PSD curve (PSD-AUC) and the max distance between data points before and after peaks of PSD (MDPSD), were calculated.

#### 2.6.1. Area under the Curve

The calculation for PSD-AUC included using the trapezoidal rule, as shown in Equation (2) [21]. In Equation (2), the $f(x_k)$ data sample represents the magnitude of PSD in Hz on the *x*-axis, $f(x_{k-1})$ represents the data sample prior to $f(x_k)$, and $\Delta(y_k) = (y_k - y_{k-1})$ represents the difference between two data samples on the *y*-axis. Once the total AUC was calculated for all speed levels, a correlation between the AUC and different speed levels for both the GAS-L and AT was performed.

$$AUC = \sum_{K-1}^{N} \left[ \frac{f(x_{k-1}) + f(x_k)}{2} \cdot \Delta y_k \right] \tag{2}$$

#### 2.6.2. Max Distance between Data Points before and after Peaks of PSD

The MDPSD was calculated following a specific procedure. Firstly, the peak point of each PSD curve was identified. Next, the number of data points was normalized to 600 samples before and after the peak through interpolation. Then, the difference between the corresponding data points before and after the peak was calculated. By generating 300 samples of these differences, the data point with the maximum difference was determined, as depicted in Figure 1. Using this method, the MDPSD for each speed level was determined, aiming to observe any distinct trends across the increasing speed levels for both muscles.

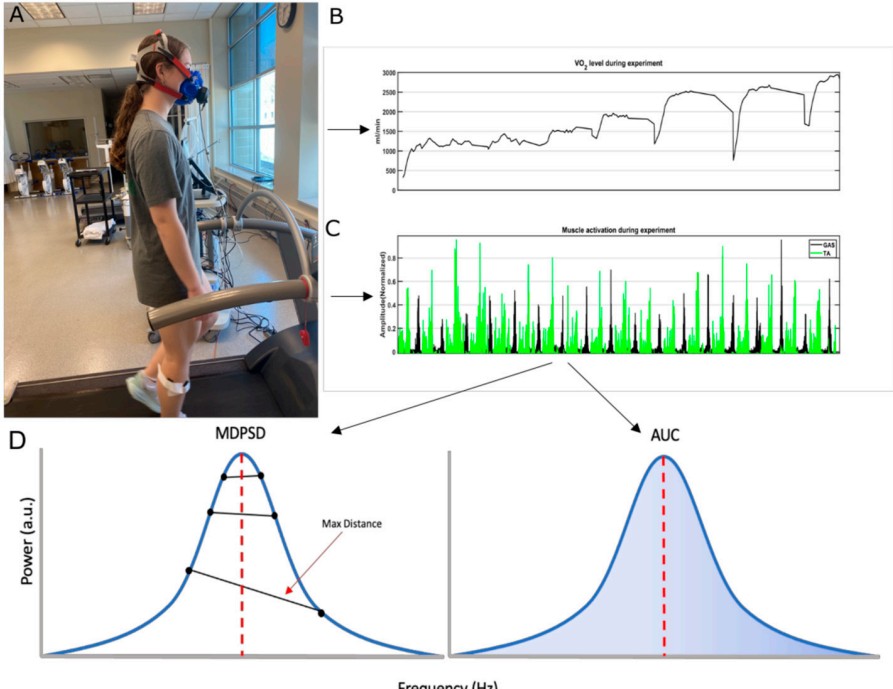

**Figure 1.** Picture (**A**) shows a participant performing the exercise protocol. Plots (**B**,**C**) provide samples of the metabolic data and EMG data that were collected during the exercise protocol. In plot (**D**), the left panel shows the distances between data points before and after the peak value. The peak value is indicated through a red vertical dashed line. The maximal distance between data points before and after peak values was used as an MDPSD measure. The MDPSD was used for further analysis. The right panel shows the AUC estimated from the PSD curve. The blue highlighted region in the PSD is the area that was used for correlation analysis.

### 2.7. Statistical Analysis

We conducted an independent *t*-test across participants to compare the mean values of the AT and GAS muscles at different speed levels. In addition, we performed an F test to examine the differences in variance for these same muscles as the speed level increased. We considered a *p*-value of $\leq 0.05$ to indicate a statistically significant difference.

## 3. Results

### 3.1. Cost of Transport at Increasing Speed Level

Tests were initially conducted to examine the impact of increasing speed levels on metabolic efficiency. COT was used as a measure to estimate the metabolic efficiency of gait. The COT also served as a reference for identifying and excluding outliers associated with running. To compare how the COT changed with increasing speed levels, correlation and trend analyses were used. COT followed a U-shaped curve, which was confirmed by fitting the COT data points with a third-order polynomial function ($R^2 = 0.98$).

Specifically, the COT was observed to be at its lowest point during the average walking speed for all participants, while it increased at both lower and higher walking speeds as shown in Figure 2. The seventh speed level represented a region where most of the participants transitioned into the running zone. Hence, the seventh speed level was removed from further analysis.

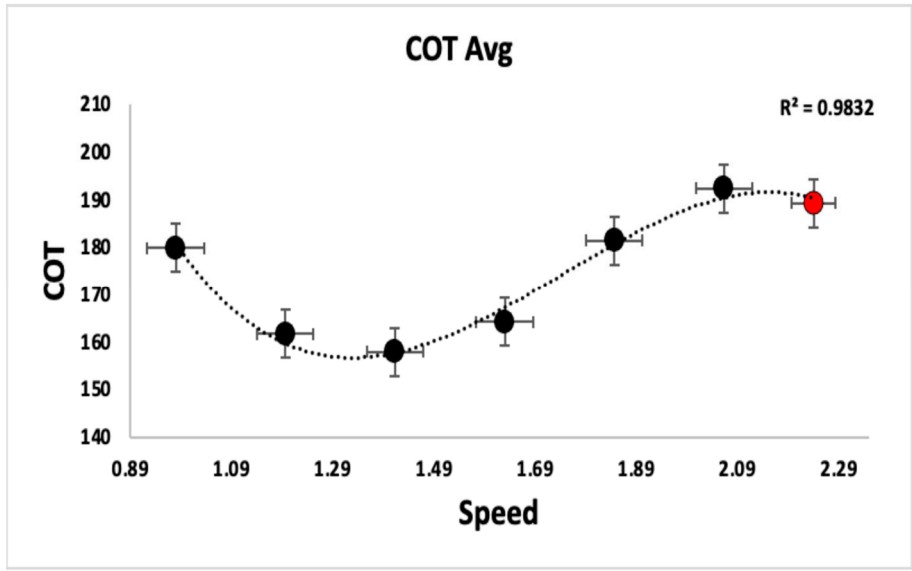

**Figure 2.** The plot shows the COT curve averaged across the participants. The standard deviations for speed are shown as horizontal error bars, and the standard deviations for COT are shown as vertical error bars. The *x*-axis is the speed in meters per second, whereas the *y*-axis is the COT in $mlO_2 \ kg^{-1} \ km^{-1}$. The red data point to the right represents the point where participants are entering into the running zone rather than walking, as the COT reduces with running.

*3.2. Muscle Activation Frequency at Incrementing Speed Levels*

The changes in the patterns of muscle activation frequency at increasing speed levels were tested, and both PSD-AUC and MDPSD were used as a measure for the analysis. The speed levels that were determined as walking ranged from 0.98 m/s to 2.06 m/s.

3.2.1. PSD-AUC at Incrementing Speed LEVEL

AT and GASL's PSD-AUC exhibited a curvilinear trend with a second-order polynomial fit (GAS-L, $R^2 = 0.93 \pm 0.07$ and AT, $R^2 = 0.77 \pm 0.31$) while walking speed levels increased. For all of the participants, GAS-L's PSD-AUC tended to increase as the walking speed increased, as shown in Figure 3B.

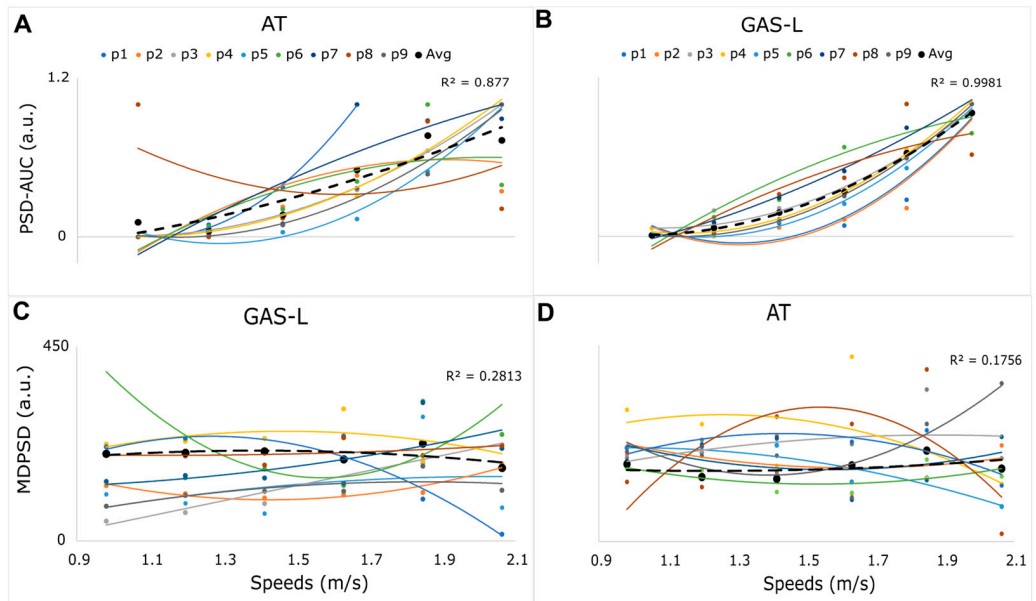

**Figure 3.** Plots (**A**,**B**) show each participant's PSD-AUC curvilinear trend for both the AT and GAS-L. The *x*-axis displays the average speed levels across participants in m/s. The *y*-axis is the normalized AUC between zero and one. Plots (**C**,**D**) show a higher nonlinear trend for GAS-L and AT, respectively. The *x*-axis displays the average speed in m/s for all the participants and the *y*-axis displays MDPSD in arbitrary units. The trends for GAS-L and AT's MDPSD are relatively more nonlinear compared to PSD-AUC across all participants. The black dashed line shows the average values of each speed level across participants.

For AT, the PSD-AUC for six of the nine participants showed a similar increasing curvilinear trend to GAS-L. However, three participants did not follow the same pattern. Two of these three participants showed a plateau at a faster walking speed, whereas one participant showed an alternating trend of high and low PSD-AUC for AT as the walking speed increased (Figure 3A). Moreover, we found statistically insignificant differences between the muscles' PSD-AUC ($p > 0.05$, independent *t*-test) at each speed level. However, as speed levels increased, we observed a statistically significant variance increase in both muscles ($p < 0.05$, f-test), with a particularly higher variance for AT compared to GAS-L, as shown in Figure 4A.

Overall, the GAS-L's PSD-AUC displayed a relatively consistent curvilinear increasing trend as speed levels were incremented, in contrast to the AT's PSD-AUC, which showed a more nonlinear and highly variable trend as speed levels increased.

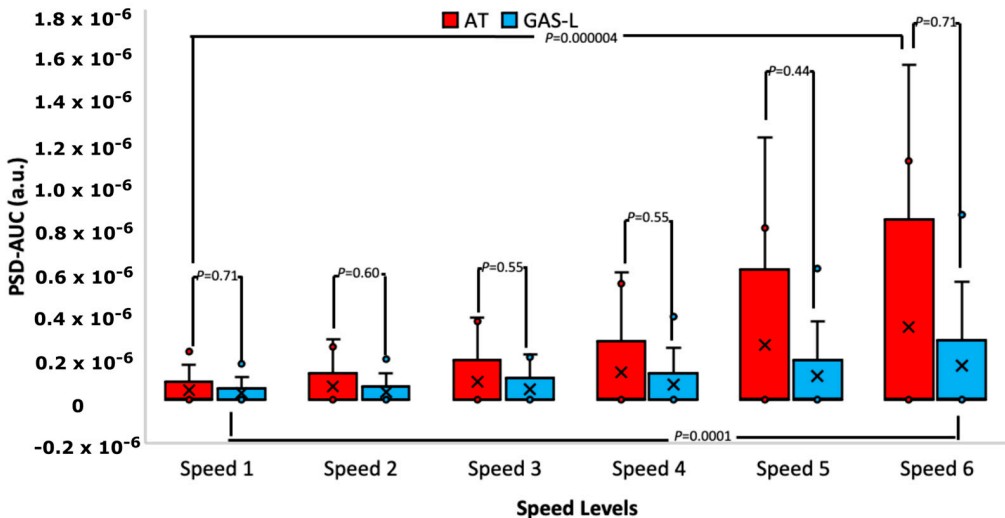

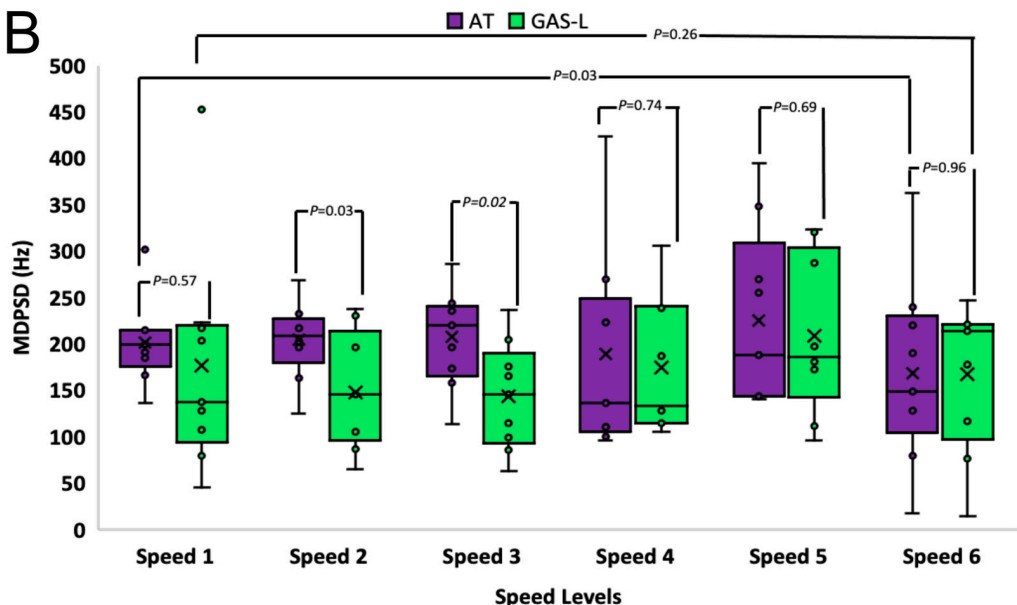

**Figure 4.** (**A**,**B**) present boxplots illustrating the PSD-AUC values and MDPSD values of both the AT and GAS-L muscles. The *x*-axis represents different speed levels ranging from 0.98 to 2.06 m/s, while the *y*-axis depicts the PSD-AUC values in arbitrary units and MDPSD values in Hz. The *p*-value resulting from the independent *t*-test conducted between the two muscles is displayed atop each speed level's boxplot, connected by lines. Furthermore, interconnected lines also indicate the *t*-test *p*-values between the slowest and fastest speed levels. In the boxplots for AT and GAS-L, a cross symbol is employed to denote the average value for each speed level, calculated across all participants.

3.2.2. MDPSD at Incrementing Speed Level

The MDPSD of AT and GAS-L showed a more nonlinear trend compared to the PSD-AUC for both muscles. This can be attributed to the low $R^2$ values obtained from the second-order polynomial fit. Specifically, the GAS-L had an average $R^2$ value of $0.28 \pm 0.32$ across participants, while the AT had an average $R^2$ value of $0.17 \pm 0.28$ across participants. Figure 3C,D show the MDPSD-GAS-L and MDPSD-AT's highly nonlinear trend.

At the average walking speed (speed level three) and speed level two, we observed a statistically significant difference ($p < 0.05$, *t*-test independent) in the frequency range

between MDPSD-AT and MDPSD-GAS-L. The frequency range for the MDPSD-GAS-L at the average walking speed was 64–237 Hz, whereas for the MDPSD-AT, it was 114–287 Hz. This suggests that the frequency range for MDPSD-GAS-L was lower compared to MDPSD-AT at slow walking speeds, as illustrated in Figure 4B. Additionally, we found that as the speed changed, the variance was statistically insignificant ($p > 0.05$).

### 3.3. Relationship between Cost of Transport and Muscle Activation Frequency at Incrementing Speed

The relationship between PSD-AUC (shown in Figure 5A,B) and COT was tested in response to the observation of more linearity in the PSD-AUC trend than MDPSD as the participants started transitioning to faster walking. A second-order polynomial fit the data points well, with an $R^2 = 0.67 \pm 0.27$ for AT and $R^2 = 0.68 \pm 0.18$ for GAS-L. These findings indicated that the lowest COT did not relate to low and high PSD-AUC values, as demonstrated in Figure 5C,D. Additionally, once the lowest COT value was achieved, the PSD-AUC tended to increase linearly with higher COT values. Therefore, it is important to note that a lower or higher PSD-AUC does not necessarily correspond to the optimal COT value.

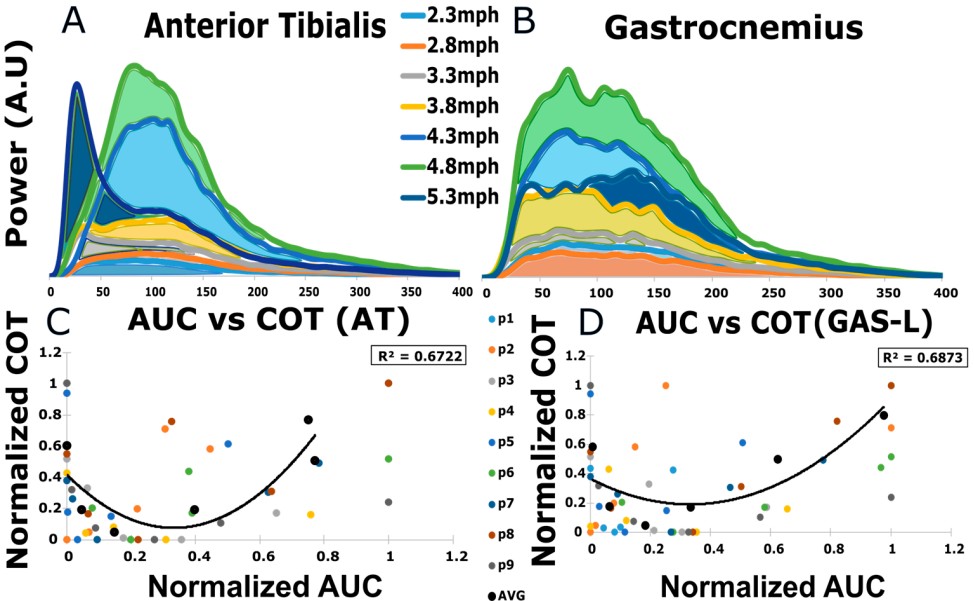

**Figure 5.** (**A**,**B**) display examples of PSD-AUC for a participant at different speed levels. The area colored differently represents different speeds. The *y*-axis is power in arbitrary units, whereas the *x*-axis is frequency in Hz. (**C**,**D**) display the relationship between normalized PSD-AUC values and normalized COT values for AT and GAS-L. The *x*-axis for PSD-AUC is normalized between 0 and 1, as is the *y*-axis for COT. The averaged $R^2$ value is enclosed in the box on the upper right side.

### 4. Discussion

We found that PSD-AUC for the AT and GAS-L showed a curvilinear increasing trend at increasing speed levels; however, the variability in those trends was higher for the AT compared to the GAS-L. We also determined that the MDPSD for the AT and GAS-L at increasing speed levels displayed more nonlinearity than the PSD-AUC. While the MDPSD did not show changes in the frequency variance as speed levels increased, the PSD-AUC demonstrated significant changes in the frequency variance as speed increased. The MDPSD also revealed individual differences in the AT (114–287 Hz) and GAS-L (64–237 Hz) frequencies at slower walking speeds, specifically at speed levels two and three, while the PSD-AUC failed to show statistically significant differences in this regard. Furthermore, the correlation analysis between COT and PSD-AUC revealed a U-shaped curve with a second-order polynomial fit. It showed that at slower walking speeds, the COT was high

due to passive dynamics, while at faster walking speeds, the COT was high due to active dynamics. This U-shaped relationship between COT and PSD-AUC suggests that a higher or lower frequency of the lower limb muscles does not guarantee efficient gait.

### 4.1. Biomechanical Inference Regarding the Change in PSD-AUC Variance and Ankle Joint Range of Motion as Speed Level Increases

Our analysis indicates a positive relationship between PSD-AUC variability and ankle joint angle kinematics for both the AT and GAS-L muscles as walking speed increases. Additionally, it shows that PSD-AUC exhibits a curvilinear increasing trend with greater consistency in the GAS-L muscle compared to the AT muscle. We suggest that the difference in ankle joint acceleration serves as a discriminating factor contributing to the highly variable curvilinear trend observed in the AT muscle when compared to the GAS-L muscle at increasing speed levels [22]. This conclusion was derived from our own findings, as well as previous studies. Research has demonstrated that the ankle joint's angular position and velocity display broader and higher ranges at varying speeds during the pre-swing phase [22], leading to increased acceleration. The heightened consistency within the GAS-L's PSD-AUC can be attributed to the regulation of this high acceleration of the ankle joint during the pre-swing phase. This regulation is crucial for effectively responding to the demands of push-off [22,23]. In contrast, the AT muscle, with its broader frequency range, regulates a narrower and lower range of ankle dorsiflexion position and velocity during the mid-swing phase [22,23], resulting in deacceleration of the ankle joint at different speed levels, setting it apart from the GAS-L muscle.

### 4.2. PSD-MDPSD Reveals Individual Differences between Muscles at Only Slow Walking Speed

While our PSD-AUC analysis revealed a statistically significant change in frequency variance as speed levels increased, it did not indicate individual differences between the GAS-L and AT. MDPSD showed a significant difference between the muscles, but only for slow walking speeds. This distinction is clear in Figure 4B, where MDPSD indicates a lower demand for GAS-L activation, and a higher demand for AT activation. The low GAS-L activation frequency could be attributed to reduced propulsive force at slow walking speeds, as a low sense of effort is required in such cases [24,25]. Additionally, the high AT muscle activation frequency at slow walking speeds (at or below average) can be attributed to proprioceptive feedback from AT, ensuring proper heel strike [26–28], suggesting the difference in AT and GAS-L muscle activation frequencies. These results further imply that high muscle fascicle stretch during increasing walking speed does not guarantee a correspondingly high muscle activation frequency, especially for the AT muscle.

Our findings indicate a higher AT activation frequency than GAS-L in terms of MDPSD. This behavior was observed between these muscles only during slow walking speeds, and not during faster walking speeds, because muscle fascicles experience more stretching during faster speeds, and this mechanical behavior has been established well in the literature for both muscles [26,29]. This suggests that AT activation is independent of fascicle stretch to a certain degree at both slow and fast walking speeds. This has also been corroborated in [26] via the use of ultrasounds, especially for slow walking. This study focused exclusively on slower walking speeds (below average). However, our study provides further evidence regarding faster walking speeds and their impact on muscle contributions, muscle activation, and energy expenditure.

### 4.3. PSD-MDPSD vs. PSD-AUC

On a comparative note, our PSD-AUC results, as compared to MDPSD, suggest that PSD-AUC is a more effective measure for capturing changes in muscle activation within the frequency domain as speed levels increase. This is attributed to PSD-AUC's ability to encompass more information under the larger area of the PSD curve, as opposed to MDPSD, which conveys information along a linear segment connecting only two data

points. However, MDPSD serves as a suitable measure for capturing differences between muscle activation frequency during slow walking speeds.

### 4.4. Energy Expenditure Relationship with Muscle Activation Frequency

When examining the correlation between COT and PSD-AUC, our study shows that there is a significant contribution of both active and passive dynamics in augmenting energy expenditure. We have explained below some of the factors that led to such an outcome.

#### 4.4.1. Slow Speed and Active-Passive Dynamic for Metabolic Cost

At slow walking speeds, we suggest that a greater demand for energy expenditure is associated more with passive dynamics than active dynamics. This increased energy expenditure, facilitated by passive dynamics, plays a crucial role in meeting the elevated requirements for maintaining proper posture and stability. The heightened demand for stability at slow walking speeds can be attributed primarily to the extended duration during which the center of gravity remains at a higher position, contrasting with the relatively shorter duration observed during faster walking speeds. As a consequence of this heightened need for stability during slow walking, compensatory responses arising from the elastic properties of muscle elements (titin filament, extracellular matrix) become more prominent [30,31]. These are initiated on both the contralateral and ipsilateral sides, leading to an elevation in the COT [32].

In addition to the mentioned muscles (AT and GAS-L), other muscles, such as the hip abductors on the ipsilateral side and the knee extensors on the contralateral side, are also recruited during slow walking, especially during single support [32]. However, due to the relatively low intensity of the exercise, it is likely that the active dynamic contribution of these muscle fibers is low due to their lower firing frequency range. It is important to note that these additional muscles were not specifically recorded in our study. Our study unequivocally demonstrates that even during low activation frequencies in the lower leg—commonly observed during slower walking speeds—there is an elevated COT, which is consistent with [33].

#### 4.4.2. Fast Speed and Active-Passive Dynamic for Metabolic Cost

During faster walking, the exercise demands placed on the body increase, necessitating greater muscle fiber recruitment, which leads to a higher demand for ATP production, thereby increasing the COT. We observed a higher contraction frequency in the GAS-L muscle (see Figure 3B), which correlated positively with increasing speed levels, especially during above average walking speeds. The progressive increase in frequency as speed levels rose above average walking speed suggests a corresponding recruitment of larger motor units, as noted in previous studies [34], leading to increased muscle contraction and force production, manifesting as a broader frequency range in a PSD of sEMG signal. Therefore, this phenomenon is an adaptive response to meet the demands of increasing exercise intensity. These findings align with the well-established concept that more motor unit recruitment is associated with a broader spectrum of firing frequencies. Overall, we found a U-shaped relationship between COT and PSD-AUC, indicating that neither excessively high nor low frequencies in lower limb muscles guarantee an efficient gait. Instead, there appears to be a specific frequency bandwidth in between these frequency ranges that governs efficient gait.

### 4.5. Limitation

The design of our study does present some limitations. One limitation of our study was the verbal cueing used for the start and stop of data collection for both the EMG and metabolic cart during each speed level. This may have led to some delay in the time stamp, as the two forms of data collection were not synced. In addition, we were also limited to examining the AT and GAS-L muscles during our study. The EMG analysis that was performed was specific to these two muscles, but the COT data collection and analysis

related to the entire body. Additionally, we did not perform kinematic, kinetic, and EMG analyses on the contralateral side. However, in the future, we plan to include such analyses in our study.

## 5. Conclusions

Our study concludes that PSD-AUC, as a measure for muscle activation frequency analysis, exhibited an increasing curvilinear trend for both muscles, with higher variability observed for the AT during incrementing speed levels. Furthermore, MDPSD displayed more nonlinearity than PSD-AUC, resulting in PSD-AUC appearing to be a better measure for muscle activation frequency analysis. However, MDPSD revealed individual differences in muscle frequencies during slow walking (AT: 114–287 Hz and GAS-L: 64–237 Hz). Most importantly, our study provides a U-shaped curve relationship between COT and PSD-AUC, similar to the speed versus COT curve. This suggests that for a low metabolic cost, neither the low nor high extremities of muscle activation frequency are preferred during gait. It is likely that both active and passive dynamics play a crucial role in maintaining this behavior.

**Author Contributions:** G.V.N., D.J.C. and R.E.S. conceived and designed the study. G.V.N., D.A. and R.E.S. collected the data. G.V.N. and R.E.S. analyzed the data. R.E.S. and G.V.N. drafted the manuscript. All authors have read and agreed to the published version of the manuscript.

**Funding:** This research received no external funding.

**Institutional Review Board Statement:** The study was conducted in accordance with the Institutional Review Board (or Ethics Committee) of Northwestern College, Orange City, Iowa (25 January 2023).

**Informed Consent Statement:** Informed consent was obtained from all participants involved in the study. Written informed consent has been obtained from the participant(s) to publish this paper.

**Data Availability Statement:** The data will be avalible on request to the corresponding author.

**Acknowledgments:** We would like to thank the Northwestern College Summer Scholarship Grant (NSG) for supporting this project.

**Conflicts of Interest:** The authors declare no conflicts of interest. The funders had no role in the design of the study; in the collection, analyses, or interpretation of data; in the writing of the manuscript; or in the decision to publish the results.

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
