# Peer review of "Muscle Activation Frequency Relationship with Cost of Transport at Increasing Walking Speed in Preliminary Study Reveals Interplay of Both Active and Passive Dynamics"

_2673-7078, doi:10.3390/biomechanics4020016_

Round 1

Reviewer 1 Report

Comments and Suggestions for Authors

In this study, authors related the EMG spectral characteristics with metabolic cost and the walking speed.

The study is clear and well written, the analyses are sound, and conclusions are supported by results, but the description of the statistical methods should be increased.

Following are some comments that I think may increase the readability of the article.

Figure 3: text is too small to be read.

Figure 5A-B: it seems that there are multiple superimposed lines that makes the figure confusing. E.g., in panel A, the dark blue curve at 5.3mph also show a green thinner curve. May you modify these panels to provide a unique line for each speed.

A ‘Statistics’ paragraph at the end of the Methods section would be helpful to better understand how the data were compared. How did you select the polynomial fit order? What is the R^2 level that you consider as an acceptable reconstruction?

In the introduction, authors assessed that amplitude measures are commonly used in literature, but a spectral analysis is more ‘physiological’ as it reflects the alteration in the firing of motor units. Despite I agree with the importance of considering the EMG spectral properties, the amplitude increase would reflect the recruitment of a larger number of motor units. Therefore, I think it would be worth to mention amplitude measures previously proposed in literature (e.g. Strazza et al., 2017, Surface-EMG analysis for the quantification of thigh muscle dynamic co-contractions during normal gait, DOI: https://doi.org/10.1016/j.gaitpost.2016.11.003; Apps et al., 2016, Lower limb joint stiffness and muscle co-contraction adaptations to instability footwear during locomotion, DOI: https://doi.org/10.1016/j.jelekin.2016.09.003; Knarr et al., 2012, Comparison of electromyography and joint moment as indicators of co-contraction, DOI: https://doi.org/10.1016/j.jelekin.2012.02.001) and what would be the outcome that authors expect from these analyses, and why they would not provide no more information with respect to the spectral analyses already implemented.

Why do the authors choose the area below the power spectral density instead of other spectral measures such as the median frequency, which is also related to muscle fatigue (Mañanas et al., 2002, Influence of estimators of spectral density on the analysis of electromyographic and vibromyographic signals, DOI: https://doi.org/10.1007/BF02347701; Borzelli et al., 2020, Identification of the best strategy to command variable stiffness using electromyographic signals, DOI 10.1088/1741-2552/ab6d88; Merletti et al., 1990, Myoelectric manifestations of fatigue in voluntary and electrically elicited contractions, DOI: https://doi.org/10.1152/jappl.1990.69.5.1810)

The affiliations of authors are not correctly reported.

Author Response

Reviewer 1

Comments and Suggestions for Authors

In this study, authors related the EMG spectral characteristics with metabolic cost and the walking speed.

The study is clear and well written, the analyses are sound, and conclusions are supported by results, but the description of the statistical methods should be increased.

Following are some comments that I think may increase the readability of the article.

Figure 3: text is too small to be read.

Dear Reviewer,

Thanks for your feedback. We have resolved this issue in the manuscript.

Figure 5A-B: it seems that multiple superimposed lines make the figure confusing. For, in panel A, the dark blue curve at 5.3mph also shows a green thinner curve. May you modify these panels to provide a unique line for each speed?

Dear Reviewer,

The color of this area is only meant to show what area under the curve is. The bottom half of the figure shows the average across the participants. We have highlighted the average values in black color.

A ‘Statistics’ paragraph at the end of the Methods section would be helpful to better understand how the data were compared. How did you select the polynomial fit order? What is the R^2 level that you consider as an acceptable reconstruction?

Dear Reviewer,

Thanks for your feedback. We have added the stats section in the manuscript.

In the introduction, the authors assessed that amplitude measures are commonly used in literature, but a spectral analysis is more ‘physiological’ as it reflects the alteration in the firing of motor units. Despite I agree with the importance of considering the EMG spectral properties, the amplitude increase would reflect the recruitment of a larger number of motor units. Therefore, I think it would be worth mentioning amplitude measures previously proposed in literature (e.g. Strazza et al., 2017, Surface-EMG analysis for the quantification of thigh muscle dynamic co-contractions during normal gait, DOI: https://doi.org/10.1016/j.gaitpost.2016.11.003; Apps et al., 2016, Lower limb joint stiffness and muscle co-contraction adaptations to instability footwear during locomotion, DOI: https://doi.org/10.1016/j.jelekin.2016.09.003; Knarr et al., 2012, Comparison of electromyography and joint moment as indicators of co-contraction, DOI: https://doi.org/10.1016/j.jelekin.2012.02.001) and what would be the outcome that authors expect from these analyses, and why they would not provide no more information concerning the spectral analyses already implemented.

Dear reviewer,

These studies are performed with a very different experimental design than ours. They only studied gait at a normal/average speed level. Our goal is not to look at the change of muscle activation frequency at one pace but at different speeds. We have added the novelty of our study in the manuscript more and have cited the relevant literature as requested. Thank you again for your feedback.

“The current research literature regarding quantifying EMG activation has certain limitations because it primarily focuses on utilizing the time domain feature at a fixed speed, such as the amplitude of EMG activity [Strazza et al., 2017], [Apps et al., 2016], [Knarr et al., 2012].”

Why do the authors choose the area below the power spectral density instead of other spectral measures such as the median frequency, which is also related to muscle fatigue (Mañanas et al., 2002, Influence of estimators of spectral density on the analysis of electromyographic and vibromyographic signals, DOI: https://doi.org/10.1007/BF02347701; Borzelli et al., 2020, Identification of the best strategy to command variable stiffness using electromyographic signals, DOI 10.1088/1741-2552/ab6d88; Merletti et al., 1990, Myoelectric manifestations of fatigue in voluntary and electrically elicited contractions, DOI: https://doi.org/10.1152/jappl.1990.69.5.1810)

Dear reviewer,

Thank you for your feedback. While we do not intend to quantify muscle fatigue in our study, our focus lies in measuring the power spectral density at various speed levels. We suggest that incorporating the area under the curve provides a more comprehensive understanding of the power spectral density curve compared to solely relying on median frequency. Thus, we have selected the area under the curve as one of our parameters. Your suggestion is intriguing, and we appreciate your insight. Since this is a preliminary study, we will consider incorporating median frequency into our future studies.

The affiliations of authors are not correctly reported.

Thank you for your feedback, we have updated the affiliation.

Reviewer 2 Report

Comments and Suggestions for Authors

General Comments

This is an interesting study, assessing  the relationship between metabolic costs and the frequency spectrum of the surface EMG signal from lateral gastrocnemius and the anterior tibialis muscles during walking at different speeds. The Authors hypothesized that walking at incrementing speed levels will have a non-linear impact on muscle activation frequencies due to an interplay of both active and passive dynamics. A total 5 men and 4 women who walked for four minutes at six different speeds (ranging from 1.8 to 5.9 mph) participated. EMG data were analyzed using 2 frequency domain features, such as Area Under the Curve of power spectral density (PSD-AUC) and the Maximal Distance Between Two Points Before and After Peak of power spectral density curve (MDPSD). Oxygen consumption was measured using open-circuit spirometry. Energy expenditure was estimated as the Cost of Transport (COT). The results of this study indicated that PSD-AUC is a better measure than MDPSD to understand the relationship between activation frequency and COT. PSD-AUC demonstrated an increasing curvilinear trend, whereas the AT displayed higher variability. The relationship between COT and PSD-AUC revealed a U-shaped curve.

The manuscript is generally well written. However, the design of this paper should be little improved before publishing.

1.    The authors should better explain the term „active and passive dynamics“ during walking. This term is regularly used in this study (in Title, Abstracts, Discussion and Conclusions). -       It has been suggested that several distinct phased of gait cycle require active control. Stable gait requires control of the body`s center of mass in relation to  its base of support. Different muscles controlled by different parts of the CNS to adapt the position of the base of support or to de-/accelerate the body center of mass. -       Passive-elastic mechanisms provide for elastic energy storage and release from terminal stance through initial swing, reducing the need for active power generation during the power burst (Whittington B, et al., The contribution of passive-elastic mechanisms to lower extremity joint kinetics during human walking. Gait and Posture, 2007, doi:10.1016/j.gaitpost.2007.08.005). This mechanism involves stretch-shortening cycle of the lower extremity extensor muscles during stance phase of gait, -       The ability of the elastic properties of passive structures to store energy has been recognized as a mechanism for lowering the energy cost of locomotion (Cavagna GA. Storage and utilization of elestic energy in skeletal muscle. Exerc Sport Sci Rev., 1977, 5: 89-129ï¼›Marsh R. L., Ellerby D. J., Carr J. A., Henry H. T. and Buchanan C. I. Partitioning the energetics of walking and running: swinging the limbs is expensive. Science, 2004, 303: 80-83. 10.1126/science.1090704;  Saibene and Minetti, 2003). -       The elastic deformation of passive structures may allow energy storage and return during gait in order to accelerate the body segments, reducing the active part (i.e. muscle contraction) of required energy. However, the extent to which this phenomenon is present during the gait remains controversial (see Koussou A et al., J Biomechanics, 2021: vol. 122). -       I suggest not to put an accent to the term “active and passive dynamics” in this manuscript. The main aim of this study was the assessment of the relationship between metabolic costs and the frequency spectrum of the surface EMG signal during walking at different speeds without any measurement of kinematic or kinetic (dynamic) parameters. The results are clearly presented and not necessary to speculate about “active and passive dynamics of gait” without any experimental assessment.       2.    They should describe more detailed the exclusion criteria of the participants in 2. Materials and Methods. For example, how was controlled that the participants: (1)  Had no metabolic syndrome (2)  Had no overweight and obesity (3)  Had no infections and signals of fatigue on assessment day. (4)  Did not take the medication or drugs (pain drugs, etc.) or caffeine during the assessment day that may influence the functional status during gait assessment. All these factors can significantly influence the results of this study.   3.    I suggest to add . 2. Materials and Methods the paragraph 2.7. Statistical analysis with description of statistical methods used in this study (with an indication of statistical significance level, p < 0.05).   4.    In my opinion, it is obligatory to add as a limiting factor of the study the facts: (1)  Small sample size (only 5 men and 4 women) with a large age range (20-48 years. i.e. young and middle-aged subjects weere mixed). (2)  Kinematic or kinetic (dynamic) parameters of gait (angles, joint torques and joint powers, or ground reaction forces) were not measured in this study. (3)  Habitual physical activity of the participants was not assessed. This notice  should be mentioned and analyzed at the end of DISCUSSION.

Specific Comments

Abstract

Page 1, line 11.

1. Please add information that 5 men and 4 women with age range 20-48 years were measured.

2. Materials and Methods

Page 3. 2. 1. Participant Recruitment and Description

Please describe more detail the exclusion criteria of the participants (see General Comments).

Page 6. Please add 2.7. Statistical Analysis (see General Comments)

4. Discussion

Page 13. 54.5. Limitation

Please add the more limiting factors of this study at the end of the Discussion (see General Comments).

Author Response

Reviewer 2

General Comments

This is an interesting study, assessing the relationship between metabolic costs and the frequency spectrum of the surface EMG signal from lateral gastrocnemius and the anterior tibialis muscles during walking at different speeds. The Authors hypothesized that walking at incrementing speed levels will have a non-linear impact on muscle activation frequencies due to an interplay of both active and passive dynamics. A total of 5 men and 4 women who walked for four minutes at six different speeds (ranging from 1.8 to 5.9 mph) participated. EMG data were analyzed using 2 frequency domain features, such as Area Under the Curve of power spectral density (PSD-AUC) and the Maximal Distance Between Two Points Before and After the Peak of the power spectral density curve (MDPSD). Oxygen consumption was measured using open-circuit spirometry. Energy expenditure was estimated as the Cost of Transport (COT). The results of this study indicated that PSD-AUC is a better measure than MDPSD to understand the relationship between activation frequency and COT. PSD-AUC demonstrated an increasing curvilinear trend, whereas the AT displayed higher variability. The relationship between COT and PSD-AUC revealed a U-shaped curve.

The manuscript is generally well-written. However, the design of this paper should be a little improved before publishing.

  1. The authors should better explain the term „active and passive dynamics“during walking. This term is regularly used in this study (in the Title, Abstracts, Discussion, and Conclusions).

Dear Reviewer,

Active control is typically associated with voluntary muscle contraction governed by the CNS, whereas passive control involves non-contractile elements like ligaments, skeletal structures, and the titin filament within muscles. Consequently, both active and passive mechanisms play a crucial role in gait control. It is hard to isolate one from another in real data. This has been thoroughly discussed in the discussion section. We also like to add that raising the center of mass and positioning it within the base of support imposes not only a burden on the contractile elements of the muscles but also on the bony structures and ligaments thus increasing the cost of transport.

Moreover, the CNS does not exclusively regulate all elements of the motor responses; rather, certain parameters are governed by the laws of physics. Latash, M. L. (2012). The bliss (not the problem) of motor abundance (not redundancy). Experimental Brain Research, 217(1), 1–5. https://doi.org/10.1007/s00221-012-3000-4.

-       It has been suggested that several distinct phases of the gait cycle require active control. A stable gait requires control of the body`s center of mass about its base of support. Different muscles are controlled by different parts of the CNS to adapt the position of the base of support or to de-/accelerate the body's center of mass. 

Dear Reviewer,

We agree with you but keeping the center of mass within the base of support is not solely the responsibility of the muscle's active elements as there are passive elements involved in that which we have explained in the previous response.

-       Passive-elastic mechanisms provide for elastic energy storage and release from the terminal stance through the initial swing, reducing the need for active power generation during the power burst (Whittington B, et al., The contribution of passive-elastic mechanisms to lower extremity joint kinetics during human walking. Gait and Posture, 2007, doi: 10.1016/j.gaitpost.2007.08.005). This mechanism involves a stretch-shortening cycle of the lower extremity extensor muscles during the stance phase of gait, 

Dear Reviewer,

We are not sure if you are making a statement, or do you have a question regarding the stretch-shortening cycle and how it relates to our study. Moreover, we are not looking at the changes in the elastic energy during the different phases of the gait in this paper. Therefore, it is hard to tell when energy has been stored and released during the gait cycle especially stretch shortening cycle. 

-       The ability of the elastic properties of passive structures to store energy has been recognized as a mechanism for lowering the energy cost of locomotion (Cavagna GA. Storage and utilization of elestic energy in skeletal muscle. Exerc Sport Sci Rev., 1977, 5: 89-129ï¼›Marsh R. L., Ellerby D. J., Carr J. A., Henry H. T. and Buchanan C. I. Partitioning the energetics of walking and running: swinging the limbs is expensive. Science, 2004, 303: 80-83. 10.1126/science.1090704;  Saibene and Minetti, 2003). 

Dear Reviewer,

Yes, it is the case when you are running. The literature that has been suggested discusses walking compared to running. However, in our study, we are not observing running. We have mentioned that when you are walking faster the high COT is attributed to the active elements of the muscles which are actin and myosin, and the passive elements are not as involved as they were during slow walking. We completely agree that when you are running there is a quick stretching and shortening of the muscles resulting in a low lack of energy.

-       The elastic deformation of passive structures may allow energy storage and return during gait to accelerate the body segments, reducing the active part (i.e. muscle contraction) of required energy. However, the extent to which this phenomenon is present during the gait remains controversial (see Koussou A et al., J Biomechanics, 2021: vol. 122). 

Dear Reviewer,

We are not proving the hypothesis regarding the elastic deformation but based on our results and data, we only observed that a high COT at slower walking cannot be attributed to high muscle activation only. Therefore, there must be factors other than the active contraction of muscle that are raising the COT. Some studies have shown that at slower walking speeds, although the hysteresis losses are higher (Zelik KE, Franz JR (2017) It’s positive to be negative: Achilles tendon work loops during human locomotion. PLoS ONE 12 (7): e0179976)

-       I suggest not putting an accent on the term “active and passive dynamics” in this manuscript. The main aim of this study was the assessment of the relationship between metabolic costs and the frequency spectrum of the surface EMG signal during walking at different speeds without any measurement of kinematic or kinetic (dynamic) parameters. The results are presented and not necessary to speculate about “active and passive dynamics of gait” without any experimental assessment.      

Dear Reviewer,

When we increase the speed of the treadmill during our experimental protocol, we are regulating the dynamics of the task. So, it is safe to consider that we are looking at muscle activation at different dynamics. We know from the literature that walking at different speed levels changes the kinetics and kinematics. Acquiring kinematic and kinetic data would have added more analysis to the research but would not have changed the results. Since this is a preliminary study, we plan to do this type of analysis in the future.

Our title also states that we are looking at a relationship between muscle activation at different speed levels and that relationship reveals, rather than validates, the existence of how active and passive dynamics are involved in gait. Our results unequivocally demonstrate that, despite the recruitment of slow-twitch fibers indicated by low activation frequency, the COT remains high. This suggests the presence of additional intrinsic factors or variables contributing to the elevated COT, which we have associated with passive dynamics in our discussion. Therefore, while speculative, it is challenging to dismiss the notion that passive dynamics play a significant role in our study, contributing to the high COT observed at slower walking speeds.

  1. They should describe more detailed the exclusion criteria of the participants in 2. Materials and Methods. For example, how was controlled that the participants:(1)  Had no metabolic syndrome (2)  Had no overweight and obesity (3)  Had no infections and signals of fatigue on assessment day. (4)  Did not take the medication or drugs (pain drugs, etc.) or caffeine during the assessment day that may influence the functional status during gait assessment. All these factors can significantly influence the results of this study.   

Dear Reviewer,

We have added a statement regarding these factors. 

“The participants in this study were neither overweight nor obese, and they exhibited no signs of metabolic syndrome, infection, or fatigue on the assessment day. Additionally, participants did not consume caffeine or prescription-based medicine on the assessment day”

  1. I suggest to add. 2. Materials and Methods paragraph 2.7. Statistical analysis with a description of statistical methods used in this study (with an indication of statistical significance level, p < 0.05).  

Dear Reviewer,

We have added a stats section as asked. 

  1. In my opinion, it is obligatory to add as a limiting factor of the study the facts:(1) a Small sample size (only 5 men and 4 women) with a large age range (20-48 years. i.e. young and middle-aged subjects were mixed). (2)  Kinematic or kinetic (dynamic) parameters of gait (angles, joint torques joint powers, or ground reaction forces) were not measured in this study. (3)  The habitual physical activity of the participants was not assessed. This notice should be mentioned and analyzed at the end of DISCUSSION.

 Dear Reviewer,

Regarding the first suggested limiting factor this is the reason we have titled our study as a preliminary study. For 2) we have updated limitations regarding kinetic and kinematic analysis, Additionally, for the third suggestion, the protocol was not extremely extensive, and each participant was a healthy individual. We have added a statement regarding the additional limiting factors.  

“Additionally, we did not perform kinematic, kinetic, and EMG analysis on the contralateral side. However, in the future, we plan to include such analysis in our study.”

Specific Comments

Abstract

Page 1, line 11.

  1. Please add information that 5 men and 4 women with age range 20-48 years were measured.

Dear reviewer, we have added the statement about participants.

  1. Materials and Methods

Page 3. 2. 1. Participant Recruitment and Description

Dear reviewer, we have added the statement about participants.

Please describe in more detail the exclusion criteria of the participants (see General Comments).

Dear reviewer we already have a statement of exclusion criteria.

“Individuals with movement disorders were not recruited in this study.”

Page 6. Please add 2.7. Statistical Analysis (see General Comments)

Dear Reviewer, we have added a stats section as asked. 

  1. Discussion

Page 13. 54.5. Limitation

Please add the more limiting factors of this study at the end of the Discussion (see General Comments).

Dear Reviewer, we have updated the limiting factors as asked. 

“Additionally, we did not perform kinematic, kinetic, and EMG analysis on the contralateral side. However, in the future, we plan to include such analysis in our study.”
